# Training-free Graph Neural Networks and the Power of Labels as Features

**Ryoma Sato**                                                            *rsato@nii.ac.jp*
*National Institute of Informatics*

**Reviewed on OpenReview:** *https://openreview.net/forum?id=7DzU88VrNU*

## Abstract

We propose training-free graph neural networks (TFGNNs), which can be used without training and can also be improved with optional training, for transductive node classification. We first advocate labels as features (LaF), which is an admissible but not explored technique. We show that LaF provably enhances the expressive power of graph neural networks. We design TFGNNs based on this analysis. In the experiments, we confirm that TFGNNs outperform existing GNNs in the training-free setting and converge with much fewer training iterations than traditional GNNs.

## 1 Introduction

Graph Neural Networks (GNNs) [12, 41] are popular machine learning models for processing graph data. GNNs show strong empirical performance in various machine learning and data mining tasks, including chemical modeling [10, 19], question answering [34, 44], and recommender systems [7, 15, 47, 49, 55].

One of the standard problem settings for GNNs is transductive node classification, where the goal is to predict the labels of the test nodes in a graph given the labels of other nodes. This setting has many applications, including document classification [20, 54], e-commerce [42, 56], and social analysis [13, 56]. Many GNNs, including Graph Convolutional Networks (GCNs) [20] and Graph Attention Networks (GATs) [45], tackled transductive node classification and showed excellent performance.

One of the challenges of GNNs is the computational cost. There are many huge graphs in practice, such as social networks and Web graphs, which contain billions of nodes. It is sometimes prohibitive to even scan the entire graph, e.g., the whole World Wide Web. Many methods to speed up GNNs have been proposed. The basic approach is sampling nodes and/or edges to reduce the graph size [4, 13, 56, 59]. In the extreme case, Sato et al. [40] proposed constant time graph neural networks by neighbor sampling. Although it drastically reduces the computational cost per iteration, it still requires many training iterations. PinSAGE [55] adopts parallel training with MapReduce as well as importance pooling to speed up the training. Although PinSAGE succeeded in training GNNs with Web-scale graphs, it requires massive computational resources. It is still challenging to instantly use GNNs with limited computational resources.

In this paper, we propose training-free graph neural networks (TFGNNs).

To design TFGNNs, we first introduce the idea of labels as features (LaF). LaF uses the node labels as features, which is admissible in the framework of transductive node classification. GNNs with LaF can utilize the label information, such as the class distribution in the neighboring nodes, to compute the node embeddings, which contain much more information than the embeddings with only the node features. We show that LaF provably enhances the expressive power of GNNs.

TFGNNs can be used without training and deployed instantly as soon as the model is initialized. This property reduces the burden of hyperparameter tuning as no training process is involved in this mode. TFGNNs can also be improved with optional training. Users can use TFGNNs with the training-free mode or train TFGNNs for few iterations when the computational resources for training are limited. This property

is useful for online learning, where training data come in a streaming manner, and the model should be updated instantly. Users can also fully train TFGNNs when the computational resources are abundant or the accuracy is required. TFGNNs enjoy the best of both worlds of nonparametric models and GNNs.

In the experiments, we confirm that TFGNNs outperform existing GNNs in the training-free setting and converge with much fewer training iterations than traditional GNNs.

The contributions of this paper are as follows:

- We advocate the use of LaF in transductive learning.

- We formally show that LaF strengthens the expressive power of GNNs.

- We proposed training-free graph neural networks (TFGNNs).

- We confirm that TFGNNs outperform existing GNNs in the training-free setting.

**Reproducibility**: Our code is available at `https://github.com/joisino/laf`.

## 2 Background

### 2.1 Notations

For every positive integer $n \in \mathbb{Z}_+$, $[n]$ denotes the set $\{1, 2, \ldots, n\}$. A graph is defined as a tuple of (i) the set $V$ of nodes, (ii) the set $E$ of edges, and (iii) the node features $\boldsymbol{X} = [\boldsymbol{x}_1, \boldsymbol{x}_2, \ldots, \boldsymbol{x}_n]^\top \in \mathbb{R}^{n \times d}$. Without loss of generality, we assume that the nodes are numbered with $1, 2, \ldots, n$. $\mathcal{Y}$ denotes the set of labels. $\boldsymbol{y}_v \in \mathbb{R}^{\mathcal{Y}}$ denotes the one-hot encoding of the label of node $v$. For every node $v \in V$, $\mathcal{N}(v)$ denotes the set of neighbors of node $v$. We adopt the numpy-like notation for indexing. For example, $\boldsymbol{X}_{:,1}$ denotes the first column of $\boldsymbol{X}$, $\boldsymbol{X}_{:,-1}$ denotes the last column of $\boldsymbol{X}$, $\boldsymbol{X}_{:,-5:}$ denotes the last five columns of $\boldsymbol{X}$, and $\boldsymbol{X}_{:,:-5}$ denotes all the columns except for the last five columns of $\boldsymbol{X}$.

### 2.2 Transductive Node Classification

**Problem (Transductive Node Classification).**
**Input**: A graph $G = (V, E, \boldsymbol{X})$, labelled nodes $V_{\text{train}} \subset V$, and node labels $\boldsymbol{Y}_{\text{train}} \in \mathcal{Y}^{V_{\text{train}}}$ of $V_{\text{train}}$.
**Output**: Predicted labels $\boldsymbol{Y}_{\text{test}} \in \mathcal{Y}^{V_{\text{test}}}$ of $V_{\text{test}} = V \setminus V_{\text{train}}$.

There are two settings for the node classification problem: transductive and inductive. In transductive node classification, one graph and the labels of some of its nodes are given, and we predict the labels of the remaining nodes. This setting uses the same graph for training and testing. This is in contrast to the inductive setting, which uses different graphs for training and testing. For example, in spam account detection, annotating spam accounts on Facebook and using the trained model on Facebook is an example of the transductive setting, while using the trained model on X (Twitter) is an example of the inductive setting.

Transductive node classification is a popular setting in the GNN community; it has been employed in well-known studies, such as GCNs [20] and GATs [45], and has been adopted in popular benchmarks such as Cora, PubMed, and CiteSeer. There are also many practical applications of transductive node classification, such as document classification [20, 54] and fraud detection[23, 24, 46].

## 2.3 Graph Neural Networks

GNNs are a popular solution for transductive node classification. We follow the message-passing framework of GNNs [10]. A message passing GNN is defined as follows:

$$h_v^{(0)} = x_v \qquad (\forall v \in V), \qquad (1)$$

$$h_v^{(l)} = f_{\text{agg}}^{(l)}(h_v^{(l-1)}, \{\!\{h_u^{(l-1)} \mid u \in \mathcal{N}(v)\}\!\}) \qquad (\forall l \in [L], v \in V), \qquad (2)$$

$$\hat{y}_v = f_{\text{pred}}(h_v^{(L)}) \qquad (\forall v \in V), \qquad (3)$$

where $f_{\text{agg}}^{(l)}$ is the aggregation function and $f_{\text{pred}}$ is the prediction head, which are typically modeled by neural networks.

## 3 LaF is Admissible, but Not Explored Well

We ask the readers to recall the problem setting of transductive node classification. We are given node labels $y_v$ of the training nodes. A typical approach for this problem is to feed node features $x_v$ for a training node $v$ to the model, predict the labels of node $v$, compute the loss based on the ground truth label $y_v$, and update the model parameters. However, how we use $y_v$ is not limited. We can use $y_v$ as features of node $v$ as well as for the loss function. This is the idea of LaF.

GNNs with LaF initialize the node embeddings in Eq. (1) as

$$h_v^{(0)} = [x_v; \tilde{y}_v] \in \mathbb{R}^{d+1+|\mathcal{Y}|}, \qquad (4)$$

where $[\cdot; \cdot]$ denotes the concatenation of vectors, and

$$\tilde{y}_v = \begin{cases} [1; y_v] & (v \in V_{\text{train}}), \\ \mathbf{0}_{1+|\mathcal{Y}|} & (v \in V_{\text{test}}), \end{cases} \qquad (5)$$

is the label vector of node $v$, and $\mathbf{0}_d$ is the zero vector of dimension $d$. LaF enables GNNs to utilize the label information, such as the class distribution in the neighboring nodes, to compute the node embeddings. Such embeddings are expected to be more informative than the embeddings without the label information. LaF is admissible in the sense that it uses only the information available in the transductive setting.

We emphasize that LaF has not been explored well in the literature on GNNs, regardless of its simplicity, with some notable exceptions [1, 50] (see Section 7 for detailed discussions). For example, GCNs [20] and GATs [45] adopt the transductive setting, and they are allowed to use the label information as features. However, they initialize the node embeddings as $h_v^{(0)} = x_v$ without using the label information. One of the contributions of this paper is that we affirm that LaF is allowed in the transductive setting.

We should be careful when training GNNs with LaF. LaF may harm the generalization performance by inducing a shortcut of copying the label feature $h_{v,d+1}^{(0)}$ to the prediction. To prevent this, we should remove the label of the center nodes in the minibatch and treat them as test nodes. Specifically, let $B \subset V_{\text{train}}$ be the set of nodes in the minibatch and we set

$$\tilde{y}_v = \begin{cases} [1; y_v] & (v \in V_{\text{train}} \setminus B), \\ \mathbf{0}_{1+|\mathcal{Y}|} & (v \in V_{\text{test}} \cup B), \end{cases} \qquad (6)$$

and predict the label $\hat{y}_v$ for $v \in B$, and compute the loss based on $\hat{y}_v$ and $y_v$. This simulates the transductive setting where the label information of the test nodes is missing, and GNNs learn how to predict the labels of the test nodes based on the label information and node features of the surrounding nodes.

## 4 LaF Strengthens the Expressive Power of GNNs

We show that LaF provably strengthens the expressive power of GNNs. Specifically, we show that GNNs with LaF can represent label propagation [58], an important model for transductive node classification, while

GNNs without LaF cannot. This result is interesting in its own right, and it also motivates the design of TFGNNs.

Label propagation is a classic method for transductive node classification. It starts random walks from a test node and outputs the label distribution of the labeled nodes the random walks first hit. The following theorem shows that GNNs with LaF can represent label propagation.

**Theorem 4.1.** *GNNs with LaF can approximate label propagation with any precision. Specifically, there exists a series of GNNs $\{f_{agg}^{(l)}\}_l$ and $f_{pred}$ such that for any positive $\varepsilon$, for any connected graph $G = (V, E, \boldsymbol{X})$, for any labeled nodes $V_{train} \subset V$ and node labels $\boldsymbol{Y}_{train} \in \mathcal{Y}^{V_{train}}$ and test node $v \in V \setminus V_{train}$, there exists $L \in \mathbb{Z}_+$ such that $l(\geq L)$-th GNN $(f_{agg}^{(1)}, \ldots, f_{agg}^{(l)}, f_{pred})$ with LaF outputs the approximation of label propagation with the error at most $\varepsilon$, i.e.,*

$$\left\|\hat{\boldsymbol{y}}_v - \hat{\boldsymbol{y}}_v^{LP}\right\|_1 \leq \varepsilon, \tag{7}$$

*where $\hat{\boldsymbol{y}}^{LP}$ is the output of label propagation for test node $v$.*

*Proof.* We prove the theorem by construction. Let

$$p_{l,v,i} \overset{\text{def}}{=} \Pr[\text{The random walk from node } v \text{ hits } V_{\text{train}} \text{ within } l \text{ steps and the first hit label is } i]. \tag{8}$$

For the labeled nodes, this is a constant, i.e.,

$$p_{l,v,i} = 1[i = \boldsymbol{y}_v] \quad (\forall l \in \mathbb{Z}_{\geq 0}, v \in V_{\text{train}}, i \in \mathcal{Y}). \tag{9}$$

For the other nodes, this can be recursively computed as follows:

$$p_{0,v,i} = 0 \quad (\forall v \in V \setminus V_{\text{train}}, i \in \mathcal{Y}), \tag{10}$$

$$
\begin{aligned}
&p_{l,v,i} \\
&= \sum_{u \in \mathcal{N}(v)} \Pr[\text{The first step is } v \to u] \cdot \Pr[\text{The random walk from node } v \text{ hits } V_{\text{train}} \text{ within} \\
&\hspace{6cm} l \text{ steps and the first hit label is } i \mid \text{The first step is } v \to u]
\end{aligned} \tag{11}
$$

$$
\begin{aligned}
&= \sum_{u \in \mathcal{N}(v)} \frac{1}{\deg(v)} \cdot \Pr[\text{The random walk from node } u \text{ hits } V_{\text{train}} \text{ within } (l-1) \text{ steps} \\
&\hspace{4cm} \text{and the first hit label is } i]
\end{aligned} \tag{12}
$$

$$= \frac{1}{\deg(v)} \sum_{u \in \mathcal{N}(v)} p_{l-1,u,i}. \tag{13}$$

These equations can be represented by GNNs with LaF. Specifically, the base case

$$p_{0,v,i} = \begin{cases} 1[i = \boldsymbol{y}_v] & (v \in V_{\text{train}}), \\ 0 & (v \in V \setminus V_{\text{train}}), \end{cases} \tag{14}$$

can be computed from $\tilde{\boldsymbol{y}}_v$ in $\boldsymbol{h}_v^{(0)}$. Let $f_{\text{agg}}^{(l)}$ always concat the first argument (i.e., $\boldsymbol{h}_v^{(l-1)}$ in Eq. (2)) to the output so that the GNN can keep the information of the input. $f_{\text{agg}}^{(l)}$ handles two cases by $\tilde{\boldsymbol{y}}_{v,1} \in \{0, 1\}$, i.e., whether $v$ is in $V_{\text{train}}$ or not. If $v$ is in $V_{\text{train}}$, $f_{\text{agg}}^{(l)}$ just outputs $1[i = \boldsymbol{y}_v]$, which can be computed by $\tilde{\boldsymbol{y}}_v$ in $\boldsymbol{h}_v^{(l-1)}$. If $v$ is not in $V_{\text{train}}$, $f_{\text{agg}}^{(l)}$ aggregates $p_{l-1,u,i}$ from $u$ in $\mathcal{N}(v)$ and takes the average, i.e., Eq. (13), which can be realized by message passing in the second argument of $f_{\text{agg}}^{(l)}$.

The final output of the GNN is $p_{l,v,i}$. The output of label propagation can be decomposed as follows:

$$\hat{\boldsymbol{y}}_{v,i}^{\mathrm{LP}}$$

$$= \Pr[\text{The first hit label is } i] \tag{15}$$

$$= \Pr[\text{The random walk from node } v \text{ hits } V_{\mathrm{train}} \text{ within } l \text{ steps and the first hit label is } i]$$
$$\quad + \Pr[\text{The random walk from node } v \text{ does not hit } V_{\mathrm{train}} \text{ within } l \text{ steps and the first hit label is } i] \tag{16}$$

$$= p_{l,v,i}$$
$$\quad + \Pr[\text{The random walk from node } v \text{ does not hit } V_{\mathrm{train}} \text{ within } l \text{ steps and the first hit label is } i] \tag{17}$$

As the second term converges to zero as $l$ increases, the GNNs approximate label propagation with any precision by increasing $l$. $\qquad\square$

We then show that GNNs without LaF cannot represent label propagation.

**Proposition 4.2.** *GNNs without LaF cannot approximate label propagation. Specifically, for any series of GNNs $\{f_{agg}^{(l)}\}_l$ and $f_{pred}$, there exists positive $\varepsilon$, a connected graph $G = (V, E, \boldsymbol{X})$, labelled nodes $V_{train} \subset V$, node labels $\boldsymbol{Y}_{train} \in \mathcal{Y}^{V_{train}}$ and test node $v \in V \setminus V_{train}$, such that for any $l$, GNN $(f_{agg}^{(1)}, \ldots, f_{agg}^{(l)}, f_{pred})$ without LaF has the error at least $\varepsilon$, i.e.,*

$$\left\| \hat{\boldsymbol{y}}_v - \hat{\boldsymbol{y}}_v^{LP} \right\|_1 \geq \varepsilon, \tag{18}$$

*where $\hat{\boldsymbol{y}}^{LP}$ is the output of label propagation for test node $v$.*

*Proof.* We construct a counterexample. Let $G$ be a cycle of four nodes. The nodes are numbered as $1, 2, 3, 4$ in the clockwise direction. All the nodes have the same node features $\boldsymbol{x}$. Let $V_{\mathrm{train}} = \{1, 2\}$ and $\boldsymbol{Y}_{\mathrm{train}} = [1, 0]^\top$. Label propagation classifies node 4 as class 1 and node 3 as class 0. However, GNNs without LaF always predict the same label for nodes 3 and 4 since they are isomorphic. Therefore, for any GNN without LaF, there is an irreducible error either for node 3 or 4. $\qquad\square$

Theorem 4.1 and Proposition 4.2 show that LaF provably enhances the expressive power of GNNs. These results indicate that GNNs with LaF are more powerful than traditional message passing GNNs such as GCNs, GATs, and GINs without LaF. Note that GINs have been considered to be the most expressive message passing GNNs, but GINs cannot represent label propagation without LaF while message passing GNNs with LaF can. This does not lead to a contradiction since the original GINs do not take the label information as input. Put differently, the input domains of the functions differ. These results indicate that it is important to consider what to input to GNNs as well as the architecture of GNNs.

## 5 Training-free Graph Neural Networks

We propose training-free graph neural networks (TFGNNs) based on the analysis in the previous section. TFGNNs can be used without training and can also be improved with optional training.

First, we define training-free models.

**Definition 5.1** (Training-free Model). We say a parametric model is training-free if it can be used without optimizing the parameters.

It should be noted that nonparametric models are training-free by definition. The real worth of TFGNNs is that it is training-free while it can be improved with optional training. Users can enjoy the best of both worlds of parametric and nonparametric models by choosing the trade-off based on the computational resources for training and the accuracy required.

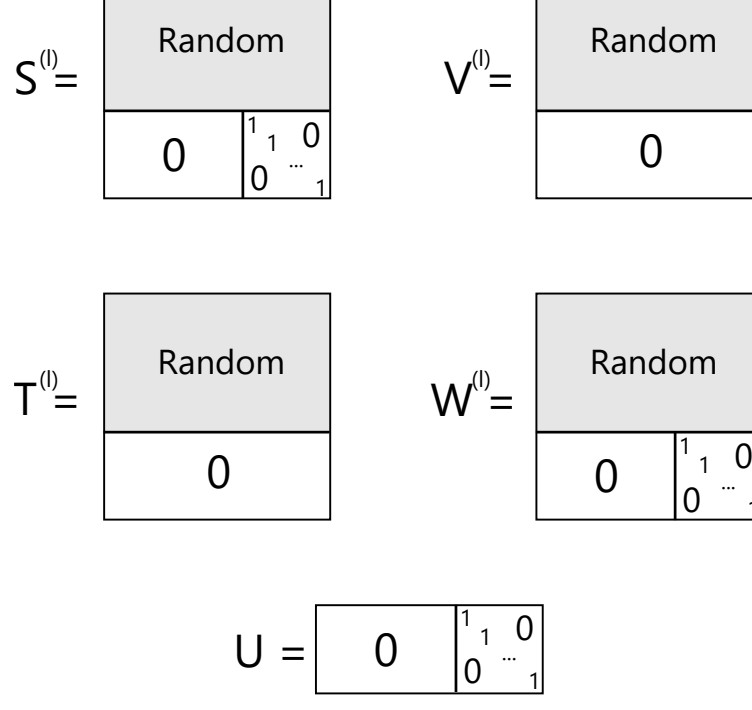

Figure 1: Initialization of TFGNNs. The parameters of the last $(1 + |\mathcal{Y}|)$ rows or $|\mathcal{Y}|$ rows are initialized by 0 or 1 in a special pattern

The core idea of TFGNNs is to embed label propagation in GNNs by Theorem 4.1. TFGNNs are defined as follows:

$$\boldsymbol{h}_v^{(0)} = [\boldsymbol{x}_v; \tilde{\boldsymbol{y}}_v] \qquad (\forall v \in V), \qquad (19)$$

$$\boldsymbol{h}_v^{(l)} = \begin{cases} \mathrm{ReLU}\left(\boldsymbol{S}^{(l)}\boldsymbol{h}_v^{(l-1)} + \frac{1}{|\mathcal{N}(v)|}\sum_{u\in\mathcal{N}(v)} \boldsymbol{V}^{(l)}\boldsymbol{h}_u^{(l-1)}\right) & (v \in V_{\mathrm{train}}, l \in [L]), \\ \mathrm{ReLU}\left(\boldsymbol{T}^{(l)}\boldsymbol{h}_v^{(l-1)} + \frac{1}{|\mathcal{N}(v)|}\sum_{u\in\mathcal{N}(v)} \boldsymbol{W}^{(l)}\boldsymbol{h}_u^{(l-1)}\right) & (v \in V_{\mathrm{test}}, l \in [L]), \end{cases} \qquad (20)$$

$$\hat{\boldsymbol{y}}_v = \mathrm{softmax}(\boldsymbol{U}\boldsymbol{h}_v^{(L)}) \qquad (\forall v \in V). \qquad (21)$$

The architecture of TFGNNs is standard, i.e., TFGNNs transform the center nodes and carry out mean aggregation from the neighboring nodes. The key to TFGNNs lies in initialization. The parameters are initialized as follows:

$$\boldsymbol{S}^{(l)}_{-(1+|\mathcal{Y}|):,:-(1+|\mathcal{Y}|)} = 0 \qquad (\forall l \in [L]), \qquad (22)$$

$$\boldsymbol{S}^{(l)}_{-(1+|\mathcal{Y}|):,-(1+|\mathcal{Y}|):} = \boldsymbol{I}_{1+|\mathcal{Y}|} \qquad (\forall l \in [L]), \qquad (23)$$

$$\boldsymbol{V}^{(l)}_{-(1+|\mathcal{Y}|):} = 0 \qquad (\forall l \in [L]), \qquad (24)$$

$$\boldsymbol{T}^{(l)}_{-(1+|\mathcal{Y}|):} = 0 \qquad (\forall l \in [L]), \qquad (25)$$

$$\boldsymbol{W}^{(l)}_{-(1+|\mathcal{Y}|):,:-(1+|\mathcal{Y}|)} = 0 \qquad (\forall l \in [L]), \qquad (26)$$

$$\boldsymbol{W}^{(l)}_{-(1+|\mathcal{Y}|):,-(1+|\mathcal{Y}|):} = \boldsymbol{I}_{1+|\mathcal{Y}|} \qquad (\forall l \in [L]), \qquad (27)$$

$$\boldsymbol{U}_{:,:-|\mathcal{Y}|} = 0, \qquad (28)$$

$$\boldsymbol{U}_{:,-|\mathcal{Y}|:} = \boldsymbol{I}_{|\mathcal{Y}|}, \qquad (29)$$

i.e., the parameters of the last $(1 + |\mathcal{Y}|)$ rows or $|\mathcal{Y}|$ rows are initialized by 0 or 1 in a special pattern (Figure 1). Other parameters are initialized randomly, e.g., by Xavier initialization [11]. The following proposition shows that the initialized TFGNNs approximate label propagation.

**Proposition 5.2.** *The initialized TFGNNs approximate label propagation. Specifically,*

$$\boldsymbol{h}^{(L)}_{v,-(|\mathcal{Y}|-i+1)} = p_{L,v,i} \tag{30}$$

*holds, where $p_{L,v,i}$ is defined in Eq. (8), and*

$$\arg\max_i \hat{\boldsymbol{y}}_{vi} = \arg\max_i p_{L,v,i} \tag{31}$$

*holds, and $p_{L,v,i} \to \hat{\boldsymbol{y}}^{LP}_{v,i}$ as $L \to \infty$.*

*Proof.* By the definitions of TFGNNs,

$$\boldsymbol{h}^{(0)}_{v,-|\mathcal{Y}|:} = \begin{cases} \boldsymbol{y}_v & (v \in V_{\text{train}}), \\ \boldsymbol{0}_{|\mathcal{Y}|} & (v \in V_{\text{test}}), \end{cases} \tag{32}$$

$$\boldsymbol{h}^{(l)}_{v,-|\mathcal{Y}|:} = \begin{cases} \boldsymbol{h}^{(l-1)}_{v,-|\mathcal{Y}|:} & (v \in V_{\text{train}}, l \in [L]), \\ \frac{1}{|\mathcal{N}(v)|} \sum_{u \in \mathcal{N}(v)} \boldsymbol{h}^{(l-1)}_{u,-|\mathcal{Y}|:} & (v \in V_{\text{test}}, l \in [L]). \end{cases} \tag{33}$$

This recursion is the same as Eqs. (9) – (13). Therefore,

$$\boldsymbol{h}^{(L)}_{v,-(|\mathcal{Y}|-i+1)} = p_{L,v,i} \tag{34}$$

holds. As $\boldsymbol{U}$ picks the last $|\mathcal{Y}|$ dimensions, and softmax is monotone,

$$\arg\max_i \hat{\boldsymbol{y}}_{vi} = \arg\max_i p_{L,v,i} \tag{35}$$

holds. $p_{L,v,i} \to \hat{\boldsymbol{y}}^{\text{LP}}_{v,i}$ as $L \to \infty$ is shown in the proof of Theorem 4.1. $\square$

Therefore, the initialized TFGNNs can be used for transductive node classification as are without training. The approximation algorithm of label propagation is seamlessly embedded in the model parameters, and TFGNNs can also be trained as usual GNNs.

## 6 Experiments

### 6.1 Experimental Setup

We use the Planetoid datasets (Cora, CiteSeer, PubMed) [54], Coauthor datasets, and Amazon datasets [42] in the experiments. We use 20 nodes per class for training, 500 nodes for validation, and the rest for testing in the Planetoid datasets following Kipf et al. [20], and use 20 nodes per class for training, 30 nodes per class for validation, and the rest for testing in the Coauthor and Amazon datasets following Shchur et al. [42]. We use GCNs [20] and GATs [45] for the baselines. We use three layered models with the hidden dimension 32 unless otherwise specified. We train all the models with AdamW [25] with learning rate 0.0001 and weight decay 0.01.

### 6.2 TFGNNs Outperform Existing GNNs in Training-free Setting

We compare the performance of TFGNNs with GCNs and GATs in the training-free setting by assessing the accuracy of the models when the parameters are initialized. The results are shown in Table 1. TFGNNs outperform GCNs and GATs in all the datasets. Specifically, both GCNs and GATs are almost random in the training-free setting, while TFGNNs achieve non-trivial accuracy. These results validate that TFGNNs

Table 1: Node classification accuracy in the training-free setting. The best results are shown in **bold**. CS: Coauthor CS, Physics: Coauthor Physics, Computers: Amazon Computers, Photo: Amazon Photo. TFGNNs outperform GCNs and GATs in all the datasets. These results indicate that TFGNNs are training-free. Note that we use three-layered TFGNNs to make the comparison fair although deeper TFGNNs perform better in the training-free setting as we confirm in Section 6.3.

|  | Cora | CiteSeer | PubMed | CS | Physics | Computers | Photo |
|---|---|---|---|---|---|---|---|
| GCNs | 0.163 | 0.167 | 0.180 | 0.079 | 0.101 | 0.023 | 0.119 |
| GCNs + LaF | 0.119 | 0.159 | 0.407 | 0.080 | 0.146 | 0.061 | 0.142 |
| GATs | 0.177 | 0.229 | 0.180 | 0.040 | 0.163 | 0.058 | 0.122 |
| GATs + LaF | 0.319 | 0.077 | 0.180 | 0.076 | 0.079 | 0.025 | 0.044 |
| TFGNNs + random initialization | 0.149 | 0.177 | 0.180 | 0.023 | 0.166 | 0.158 | 0.090 |
| TFGNNs (proposed) | **0.600** | **0.362** | **0.413** | **0.601** | **0.717** | **0.730** | **0.637** |

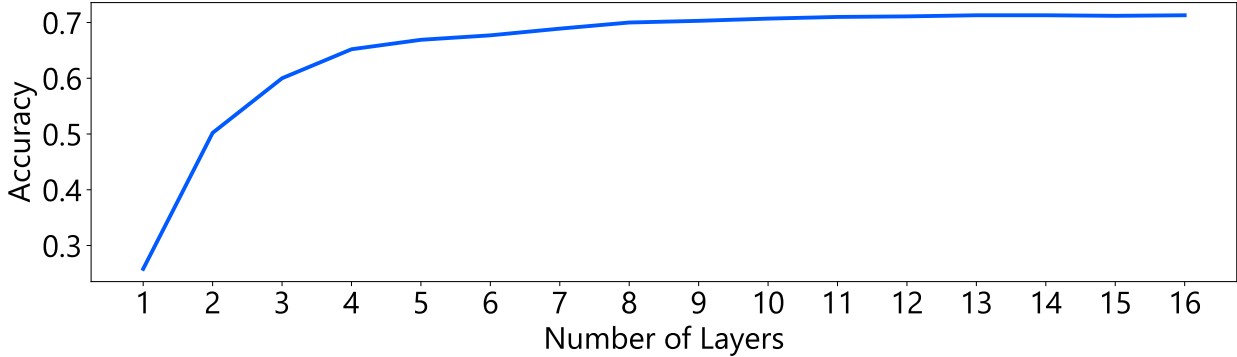

Figure 2: Deep TFGNNs perform better in the training-free setting. The x-axis is the number of layers, and the y-axis is the accuracy of the models for the Cora dataset in the training-free setting. These results show that deeper TFGNNs perform better in the training-free setting.

meet the definition 5.1 of training-free models. We can also observe that GCNs, GATs, and TFGNNs do not benefit from LaF in the training-free settings if randomly initialized. These results indicate that both LaF and the initialization of TFGNNs are important for training-free performance.

### 6.3 Deep TFGNNs Perform Better in Training-free Setting

We confirm that deeper TFGNNs perform better in the training-free setting. We have used three-layered TFGNNs so far to make the comparison fair with existing GNNs. Proposition 5.2 shows that the initialized TFGNNs converge to label propagation as the depth goes to infinity, and we expect that deeper TFGNNs perform better in the training-free setting. Figure 2 shows the accuracy of TFGNNs with different depths for the Cora dataset. We can observe that deeper TFGNNs perform better in the training-free setting until the depth reaches around 10, where the performance saturates. It is noteworthy that GNNs have been known to suffer from the oversmoothing problem [22, 33], and the performance of GNNs degrades as the depth increases. It is interesting that TFGNNs do not suffer from the oversmoothing problem in the training-free setting. It should be noted that it does not necessarily mean that deeper models perform better in the optional training mode because the optional training may break the structure introduced by the initialization of TFGNNs and may lead to oversmoothing and/or overfitting. We leave it as a future work to overcome these problems by adopting countermeasures such as initial residual and identity mapping [5], MADReg [3], and DropEdge [35].

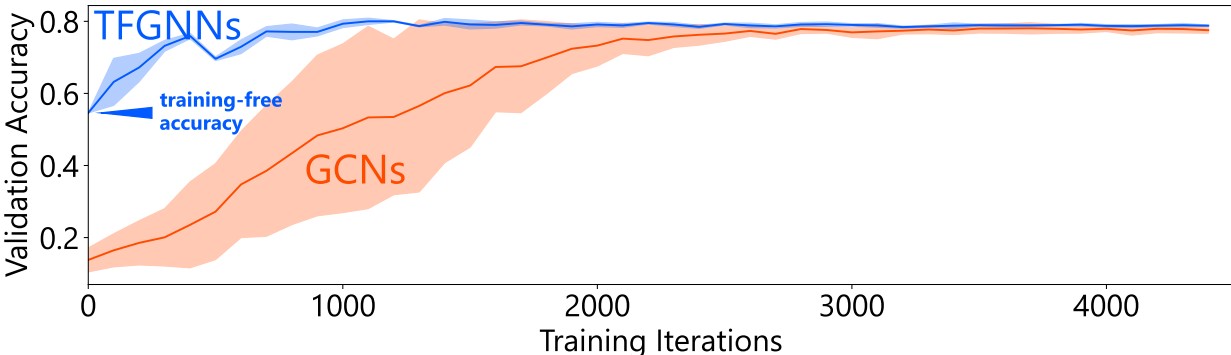

Figure 3: TFGNNs converge fast. The x-axis is the number of training iterations, and the y-axis is the validation accuracy of the models for the Cora dataset. These results show that TFGNNs in the optional training mode converge much faster than GCNs.

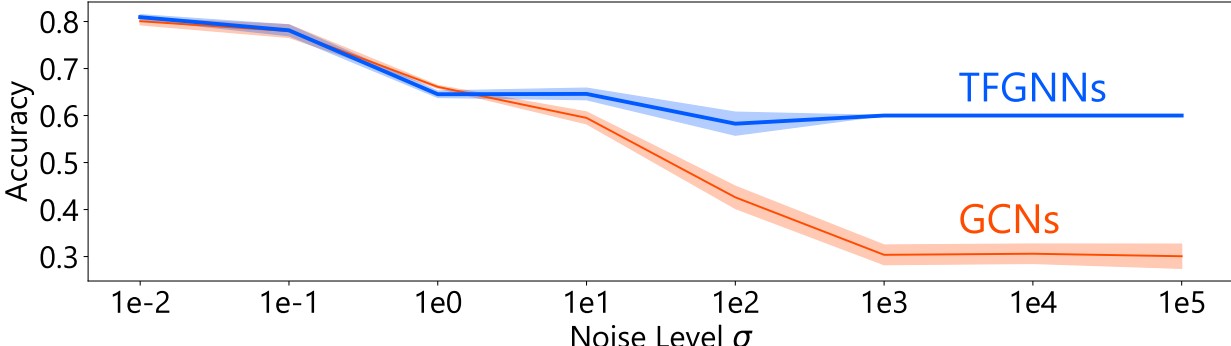

Figure 4: TFGNNs are robust to feature noise. The x-axis is the standard deviation of the Gaussian noise added to the node features, and the y-axis is the accuracy of the models for the Cora dataset. Both models are trained. These results show that TFGNNs are more robust to feature noise than GCNs.

### 6.4 TFGNNs Converge Fast

In the following, we investigate the optional training mode of TFGNNs. We train the models with three random seeds and report the average accuracy and standard deviation. We use baseline GCNs without LaF (i.e., the original GCNs) as the baseline.

First, we confirm that TFGNNs in the optional training mode converge faster than GCNs. We show the training curves of TFGNNs and GCNs for the Cora dataset in Figure 3. TFGNNs converge much faster than GCNs. We hypothesize that TFGNNs converge faster because the initialized TFGNNs are in a good starting point, while GCNs start from a completely random point and require many iterations to reach a good point. We can also observe that fully trained TFGNNs perform on par with GCNs. These results indicate that TFGNNs enjoy the best of both worlds: TFGNNs perform well without training and can be trained faster with optional training.

### 6.5 TFGNNs are Robust to Feature Noise

As TFGNNs use both node features and label information while traditional GNNs rely only on node features, we expect that TFGNNs are more robust to feature noise than traditional GNNs. We confirm this in this section. We add i.i.d. Gaussian noise with standard deviation $\sigma$ to the node features and evaluate the accuracy of the models. We train TFGNNs and GCNs with the Cora dataset. The results are shown in Figure 4. TFGNNs are more robust to feature noise especially in high noise regimes where the performance

of GCNs degrades significantly. These results indicate that TFGNNs are more robust to i.i.d. Gaussian noise to the node features than traditional GNNs.

## 7 Related Work

### 7.1 Labels as Features and Training-free GNNs

The most relevant work is Wang et al. [50], who proposed to use node labels in GNNs. This technique was also used in Addanki et al. [1] and analyzied in Wang et al. [51]. The underlying idea is common with LaF, i.e., use of label information as input to transductive GNNs. A similar result as Theorem 4.1 was also shown in [51]. However, the focus is different, and there are different points between this work and theirs. We propose the training-free + optional training framework for the first time. The notable characteristics of GNNs are (i) TFGNNs receive both original features and LaF, (ii) TFGNNs can be deployed without training, and (iii) TFGNNs can be improved with optional training. Besides, we provide detailed analysis and experiments including the speed of convergence and noise robustness. Our results provide complementary insights to the existing works.

Another related topic is graph echo state networks [8, 9, 29], which lead to lightweight models for graph data. The key idea is to use randomly initialized fixed weights for aggregation. The main difference is that graph echo state networks still require to train the output layer, while TFGNNs can be used without training. These methods are orthogonal, and it is an interesting direction to combine them to further improve the performance.

### 7.2 Speeding up GNNs

Various methods have been proposed to speed up GNNs to handle large graph data. GraphSAGE [13] is one of the earliest methods to speed up GNNs. GraphSAGE employs neighbor sampling to reduce the computational cost of training and inference. It samples a fixed number of neighbors for each node and aggregates the features of the sampled neighbors. An alternative sampling method is layer-wise sampling introduced in FastGCN [4]. Huang et al. [16] further improved FastGCN by using an adaptive node sampling technique to reduce the variance of estimators. LADIES [59] combined neighbor sampling and layer-wise sampling to take the best of both worlds. Another approach is to use smaller training graphs. ClusterGCN [6] uses a cluster of nodes as a mini-batch. GraphSAINT [56] samples subgraphs by random walks for each mini-batch.

It should also be noted that general techniques to speed up neural networks, such as mixed-precision training [30], quantization [17, 21, 43, 48, 52], and pruning [2, 14] can be applied to GNNs.

These methods mitigate the training cost of GNNs, but they still require many training iterations. In this paper, we propose training-free GNNs, which can be deployed instantly as soon as the model is initialized. Besides, our method can be improved with optional training. In the optional training mode, the speed up techniques mentioned above can be combined with our method to reduce the training time further.

### 7.3 Expressive Power of GNNs

Expressive power (or representation power) means what kind of functional classes a model family can realize. The expressive power of GNNs is an important field of research in its own right. If GNNs cannot represent the true function, we cannot expect GNNs to work well however we train them. Therefore, it is important to elucidate the expressive power of GNNs. Originally, Morris et al. [31] and Xu et al. [53] showed that message-passing GNNs are at most as powerful as the 1-WL test, and they proposed $k$-GNNs and GINs, which are as powerful as the $k$-(set)WL and 1-WL tests, respectively. GINs are the most powerful message-passing GNNs. Sato [38, 39] and Loukas [26] showed that message-passing GNNs are as powerful as a computational model of distributed local algorithms, and they proposed GNNs that are as powerful as port-numbering and randomized local algorithms. Loukas [26] showed that GNNs are Turing-complete under certain conditions (i.e., with unique node ids and infinitely increasing depths). Some other works showed that GNNs can solve or cannot solve some specific problems, e.g., GNNs can recover the underlying geometry [37], GNNs cannot

recognize bridges and articulation points [57]. There are various efforts to improve the expressive power of GNNs by non-message-passing architectures [27, 28, 32]. We refer the readers to survey papers [18, 36] for more details on the expressive power of GNNs.

We contributed to the field of the expressive power of GNNs by showing that GNNs with LaF are more powerful than GNNs without LaF. Specifically, we showed that GNNs with LaF can represent an important model, label propagation, while GNNs without LaF cannot. It should be emphasized that GINs, the most powerful message-passing GNNs, and Turing-complete GNNs cannot represent label propagation without LaF because they do not have access to the label information label propagation uses, and also noted that GINs traditionally do not use LaF. This result indicates that it is important to consider what to input to the GNNs as well as the architecture of the GNNs for the expressive power of GNNs. This result provides a new insight into the field of the expressive power of GNNs.

## 8 Limitations

Our work has several limitations. First, LaF and TFGNNs cannot be applied to inductive settings while most GNNs can. We do not regard this as a negative point. Popular GNNs such as GCNs and GATs are applicable to both transductive and inductive settings and are often used for transductive settings. However, this also means that they do not take advantage of transductive-specific structures (those that are not present in inductive settings). We believe that it is important to exploit inductive-specific techniques for inductive settings and transductive-specific techniques (such as LaF) for transductive settings in order to pursue maximum performance.

Second, TFGNNs cannot be applied to heterophilious graphs, or its performance degrades as TFGNNs are based on label propagation. The same argument mentioned above applies. Relying on homophilious graphs is not a negative point in pursuing maximum performance. It should be noted that LaF may also be exploited in heterophilious settings as well. Developing training-free GNNs for heterophilious graphs based on LaF is an interesting future work.

Third, we did not aim to achieve the state-of-the-art performance. Exploring the combination of LaF with fancy techniques to achieve state-of-the-art performance is left as future work.

Finally, we did not explore applications of LaF other than TFGNNs. LaF can help other GNNs in non-training-free settings as well. Exploring the application of LaF to other GNNs is left as future work.

## 9 Conclusion

In this paper, we made the following contributions.

- We advocated the use of LaF in transductive learning (Section 3).
  - We confirmed that LaF is admissible in transductive learning, but LaF has not been explored in the field of GNNs such as GCNs and GATs.
- We formally showed that LaF strengthens the expressive power of GNNs (Section 4).
  - We showed that GNNs with LaF can represent label propagation (Theorem 4.1) while GNNs without LaF cannot (Proposition 4.2).
- We proposed training-free graph neural networks, TFGNNs (Section 5).
  - We showed that TFGNNs defined by Eqs. (19) – (29) meet the requirements of training-free models (Definition 5.1) by showing that the initialized TFGNNs approximate label propagation in Proposition 5.2.
- We confirmed that TFGNNs outperform existing GNNs in the training-free setting. (Section 6)
  - We showed that TFGNNs outperform GCNs and GATs in all of the seven datasets in the training-free setting (Table 1).

– TFGNNs achieve non-trivial accuracy without training and can be deployed instantly as soon as the model is initialized. These results corroborate that TFGNNs are training-free (Definition 5.1).

We also note that our idea can be applied to other machine learning models than graph neural networks. We hope that this papers opens the door to a new research direction of training-free neural networks.

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
