# OpenReview forum: "Training-free Graph Neural Networks and the Power of Labels as Features"
_TMLR — Accepted by TMLR_

### Review · Reviewer_9hwo · 2024-05-24

**Summary Of Contributions:**

In this artice, the authors propose a new methodology for performing transductive node classification within a graph, where the task is to classify some nodes of a given graph, given some other nodes (on the same graph) on which the labels are known. For this, the authors suggest to use labels as features (LaF), that is, to encode the ground-truth labels (of the training nodes) as explicit features for message-passing graph neural network (GNN) architectures, instead of only using external features for the GNNs and then inferring the labels from the GNNs outputs. On the theoretical side, the authors demonstrate that, upon careful initialization of the GNN weights, and even without any training, the GNNs fed with LaF are at least as powerful as label propagation, where labels are inferred from the first training nodes reached by random walks originating from the test node to classify. On the experimental side, the authors designed experiments where untrained message-passing GNNs with LaF and clever initialization achieved better accuracies than other untrained GNN architectures (even with LaF). They also empirically showed faster convergence to the saturating accuracies, as well as better robustness to external feature noise.

**Audience:**

Yes

**Claims And Evidence:**

Yes

**Requested Changes:**

p3, first paragraph in section 3: the vector y_v has not beed defined. I figured from the text that it must be some sort of one-hot encoding of the labels, but this should be made explicit.

p5, last line in the proof of theorem 4.1: the second term has to converge to zero (the result is not true if this term only diminishes, as it could decrease to a minimum other than zero). Furthermore, I think this convergence has to be uniform across all nodes for the result to hold in the case of graphs with infinitely many nodes (if the graph is assumed to be finite, simple convergence is enough).

p6: a complementary figure could definitely be useful to illustrate equations (22) to (28).

p6: the role of the matrix U is unclear to me. What are the coefficients of the columns 0 to -|Y|-1 (in numpy notations)? If they are nonzero, then I don't see why "U picks the last |Y| dimensions" (as claimed at the end of the page). If they are zero, then 1. it should be mentioned explicitly, 2. the external features (other than the labels) are not used at all when building the predictions.

p7, table 1: why are there no variances in the results? It would be more convincing to make several runs with different training nodes, and provide the average + standard deviations of the accuracies. Also, it would be nice to explain / comment on the fact that for other GNN architectures, adding LaF actually degrades the accuracy (such as, e.g., GAT in CiteSeer), which looks counter-intuitive.

p8, Figure 1: why is there no variance in the curve whereas there is variance in Figures 2 and 3?

p8, Sections 6.4 and 6.5: were the GCNs trained with or without LaF? Providing both (and thus making three curves in Figs 2 and 3; one for TFGNNs, one for GCNs without LaF, one for GCNs with LaF) would be better.

**Strengths And Weaknesses:**

This article is clearly written, tackles an important problem, and provide an interesting approach, which is both easy to deploy and backed up by theoretical guarantees in the context of transductive node classification. To me, the main weaknesses are about some imprecisions in the text, as well as some missing details and results in the experiments. Overall, I think it is well suited for TMLR after some clarifications have been made (see below).

---

> ### Author Response · Authors · 2024-07-05
>
> We appreciate the detailed review.
>
> > p3, first paragraph in section 3: the vector y_v has not beed defined. I figured from the text that it must be some sort of one-hot encoding of the labels, but this should be made explicit.
>
> Thank you for pointing this out. We have added the definition of y_v in the notation section.
>
> > p5, last line in the proof of theorem 4.1: the second term has to converge to zero (the result is not true if this term only diminishes, as it could decrease to a minimum other than zero). Furthermore, I think this convergence has to be uniform across all nodes for the result to hold in the case of graphs with infinitely many nodes (if the graph is assumed to be finite, simple convergence is enough).
>
> We have fixed this to converge to zero in the revision. We assume the number of nodes is $n$, which is finite.
>
> > p6: a complementary figure could definitely be useful to illustrate equations (22) to (28).
>
> Thank you for the suggestion. We have added a figure to illustrate the equations in the revision.
>
> > p6: the role of the matrix U is unclear to me. What are the coefficients of the columns 0 to -|Y|-1 (in numpy notations)? If they are nonzero, then I don't see why "U picks the last |Y| dimensions" (as claimed at the end of the page). If they are zero, then 1. it should be mentioned explicitly, 2. the external features (other than the labels) are not used at all when building the predictions.
>
> The coefficients of the columns 0 to -|Y|-1 are zero. We have clarified this in the revision. The external features are not used at initialization, but they become nonzero in the optional training phase.
>
> > p8, Figure 1: why is there no variance in the curve whereas there is variance in Figures 2 and 3?
>
> We used the standard split (defined in the Planetoid dataset), which is deterministic. In the training-free setting, there are no variance due to training. The parameters relevant to outputs in TFGNNs are initialized by Eqs. (22) - (28), so there are no variance due to the initialization. Figures 2 and 3 involve training and thus have variance introduced by the stochasticity of training (such as sample order and the random initialization part that is suppressed in the initialized model).
>
> > p7, table 1: why are there no variances in the results? It would be more convincing to make several runs with different training nodes, and provide the average + standard deviations of the accuracies. Also, it would be nice to explain / comment on the fact that for other GNN architectures, adding LaF actually degrades the accuracy (such as, e.g., GAT in CiteSeer), which looks counter-intuitive.
>
> We have conducted experiments with 3 different seeds. Note that there are no variances for TFGNNs and there are only variances due to initialization for other models in the training-free settings. The tendency is the same as in the original experiments.
>
> | Method | Cora | CiteSeer | PubMed | CS | Physics | Computers | Photo |
> |------|------|------|------|------|------|------|------|
> | GCNs | 0.133 ± 0.023 | 0.162 ± 0.003 | 0.258 ± 0.110 | 0.061 ± 0.013 | 0.247 ± 0.162 | 0.045 ± 0.016 | 0.200 ± 0.059 |
> | GCNs + LaF | 0.213 ± 0.075 | 0.156 ± 0.061 | 0.333 ± 0.108 | 0.077 ± 0.042 | 0.137 ± 0.026 | 0.137 ± 0.115 | 0.108 ± 0.041 |
> | GATs | 0.131 ± 0.010 | 0.183 ± 0.034 | 0.333 ± 0.108 | 0.179 ± 0.070 | 0.110 ± 0.027 | 0.087 ± 0.031 | 0.141 ± 0.027 |
> | GATs + LaF | 0.234 ± 0.121 | 0.150 ± 0.052 | 0.260 ± 0.114 | 0.067 ± 0.039 | 0.251 ± 0.185 | 0.065 ± 0.035 | 0.073 ± 0.021 |
> | TFGNNs + random initialization | 0.143 ± 0.009 | 0.116 ± 0.044 | 0.256 ± 0.107 | 0.040 ± 0.023 | 0.258 ± 0.178 | 0.115 ± 0.061 | 0.147 ± 0.081 |
> | TFGNNs (proposed) | **0.600 ± 0.000** | **0.362 ± 0.000** | **0.413 ± 0.000** | **0.601 ± 0.000** | **0.717 ± 0.000** | **0.730 ± 0.000** | **0.637 ± 0.000** |
>
> The reason why adding LaF sometimes degrades the accuracy is not clear. Most of them are due to variance. We have not found other reasons yet.
>
> > p8, Sections 6.4 and 6.5: were the GCNs trained with or without LaF? Providing both (and thus making three curves in Figs 2 and 3; one for TFGNNs, one for GCNs without LaF, one for GCNs with LaF) would be better.
>
> The baseline GCNs were trained without LaF (as usual GCNs). We have clarified this in the revision.

---

### Review · Reviewer_KtFe · 2024-05-26

**Summary Of Contributions:**

1. The authors advocate the labels as features for transductive learning. And they formally demonstrate the LaF can strengthen the expressive power of GNN.
2. They propose the TFGNN based on the LaF technique, which is training-free and can be promoted by training as well.

**Audience:**

Yes

**Claims And Evidence:**

Yes

**Requested Changes:**

1. Please refer to Weakness 1. Please add explanations of why such initialization works.
2. Please refer to Weakness 2. Please add suggested experimental results on both training-free and training setting to make it more convincing.
3. Please refer to Weakness 3. Please demonstrate the value of "training-free" of your proposed method.
4. Please refer to Weakness 4. Please add experiments and discussion on scenario with sparse labels.
5. Please refer to Weakness 5. Please add experiments on training and inference latency.

**Strengths And Weaknesses:**

Strengths:
1. The label as feature technique is simple but seems effective with experimental results.
2. The effect of LaF are sufficiently proven.
3. The TFGNN shows impressive performance compared with existing baselines in a training-free setting.

Weakness:
1. It seems the initialization of TFGNN is an important technique. But it lacks explanation of why such initialization works.
2. It lacks sufficient comparison of TFGNN and other GNN baselines when conducting optimal training. Furthermore, in training-free setting, the baselines is not sufficient. For example, what will happen if you replace LaF to other features such as random feature or laplacian eigen-vectors?
3. TFGNN is limited in transudative learning and in a semi-supervised learning setting. In my opinion, the training-free benefits come from the following aspects (not limited to): 1) annotation is expensive, 2) training large graph is expensive. However, the proposed TFGNN is relied on the annotations (they use labels as features). Furthermore, the scalability of TFGNN is not demonstrated when graph is very large. Although this is an interesting work, the value of "training-free" is limited.
4. TFGNN is heavily relied on the labels. But in reality, there are many scenario that the labels are sparse. Is this method robust to these scenarios?
5. The concerns of training and inference latency. This method needs to sample subgraphs for each target node, thus it has low efficiency.

---

> ### Author Response · Authors · 2024-07-05
>
> Thank you for the detailed review.
>
> > It seems the initialization of TFGNN is an important technique. But it lacks explanation of why such initialization works.
>
> This initialization makes TFGNN simulate label propagation as soon as the model is initialized as we proved in Theorem 4.1.
>
> > Please add suggested experimental results on both training-free and training setting to make it more convincing.
>
> We conducted additional experiments with random features with the same dimension as the proposed LaF. The results are as follows:
>
> Training-free setting (corresponding to Section 6.2)
> | Method | Cora | CiteSeer | PubMed | CS | Physics | Computers | Photo |
> |------|------|------|------|------|------|------|------|
> | GCNs + random features | 0.186 | 0.224 | 0.18 | 0.038 | 0.164 | 0.060 | 0.120 |
> | GATs + random features | 0.319 | 0.077 | 0.18 | 0.076 | 0.079 | 0.025 | 0.044 |
> | TFGNNs (proposed) | **0.600** | **0.362** | **0.413** | **0.601** | **0.717** | **0.730** | **0.637** |
>
> Optional-training setting (corresponding to Section 6.4)
> | Method | Iter 0 | Iter 100 | Iter 200 | Iter 300 | Iter 400 | Iter 500 | Iter 1000 | Iter 2000 | Iter 3000 | Iter 4000 |
> |------|------|------|------|------|------|------|------|------|------|------|
> | GCNs + random features | 0.162 | 0.176 | 0.23 | 0.288 | 0.31 | 0.31 | 0.336 | 0.65 | 0.76 | 0.762 |
> | TFGNNs (proposed) | **0.546** | **0.724** | **0.69** | **0.736** | **0.76** | **0.69** | **0.784** | **0.784** | **0.79** | **0.786** |
>
> These results show that the proposed TFGNN is more effective than random features in both training-free and optional training settings.
>
> > Please refer to Weakness 3. Please demonstrate the value of "training-free" of your proposed method.
>
> The number of required labels is the same as in standard GNNs (such as GCNs). Thus the cost of labeling is the same. However, the proposed LaF can skip the training phase, and thus the computational cost and the burden of hyperparameter tuning are reduced.
>
> > Please refer to Weakness 4. Please add experiments and discussion on scenario with sparse labels.
>
> We use only 20 labels per class in all the experiments. For example, only < 5% of nodes are labeled in the Cora dataset, and < 0.5% of nodes are labeled in the Pubmed dataset. TFGNNs work under this sparsity.
>
> > The concerns of training and inference latency. This method needs to sample subgraphs for each target node, thus it has low efficiency.
>
> Standard GNNs also need to sample subgraphs (ego-networks) to conduct mini-batch training and inference. Therefore, the efficiency of TFGNNs is the same as that of standard GNNs. The difference is that TFGNNs cannot be trained (in the optional training mode) in a full-batch manner.

---

### Review · Reviewer_uEBa · 2024-06-27

**Summary Of Contributions:**

The authors propose the somewhat strange yet defensible idea of graph neural networks which work without any training. The idea is that in many situations, graph neural networks are used for transductive node classification, where the goal is to take a graph and impute information about neighboring nodes from some known nodes on the same graph. In particular, the authors point out that it is completely admissible in this setting to treat the label information on the known nodes as feature information (this is not “training on the test set”, since the labels of the target nodes are still unknown). The authors then show the somewhat straightforward fact that with labels as features, transductive GNNs can implement the classic algorithm of label propagation, while in general they cannot without labels as features. Finally, they describe their training-free graph neural networks (TFGNN). These are initialized to approximate label propagation and can therefore be immediately used, without any training, to perform transductive node classification. But they are also just regular graph neural networks which can still be trained for improved performance. In experiments on small standard datasets, the authors show that TF-GNNs outperform other GNNs in a number of interesting ways.

**Audience:**

Yes

**Broader Impact Concerns:**

I don't see any broader impact concerns.

**Claims And Evidence:**

Yes

**Requested Changes:**

Comments for authors: the discussion around equation 6 is a little unclear to me still. My sense is that doing things this way is necessary for good generalization, but that it would not actually be incorrect to simply train on the labels as features. If so, I think that should be made explicit, as “leaking information” sounds like something which would make any reported results invalid, rather than just being something that happens during training and hurts performance. [non-critical change]

Subsections 6.5 and 6.6:

These are reasonable experiments, which are suggestive evidence for the benefits of TF-GNNs, but I think the claims made in the text are too strong given the experiments. I would ask the authors to simply add some small amount of hedging and qualifying language to acknowledge the more limited scope of the experiments, which should not be difficult. I have nonetheless answered "Yes" to the claims and evidence question on the review form, because I think these changes should be pretty small [critical change].

**Strengths And Weaknesses:**

Strengths:

The idea is simple and elegant. It is surprising that this obvious fact was not used before, so it's good for the authors to illustrate it.

The paper is admirably clear.

The approach does seem to work well, at least on the small examples tried.

Overall I see why it has been submitted to TMLR, as it is clearly an interesting and useful contribution, yet one which might be rejected from other conferences for being "too simple" (in reality, a strength).

Weaknesses:

- The experimental section is a good demonstration of the approach, but it’s still relatively limited. I think the claims made aren't completely supported by the experiments. See requested changes, I think this is easily fixable.

- I did not get any deep insight from the proof about label propagation (it seems to just point out the obvious fact that one can’t propagate labels if one doesn’t know the labels).

- This is more a weakness of me as a reviewer, but I don't know whether or not there is some work out in the broader graph-machine-learning (not just graph neural network) literature that would make this approach less novel or less interesting.

---

> ### Author Response · Authors · 2024-07-05
>
> Thank you for the detailed review. We answer your concerns and requests.
>
> > Comments for authors: the discussion around equation 6 is a little unclear to me still. My sense is that doing things this way is necessary for good generalization, but that it would not actually be incorrect to simply train on the labels as features. If so, I think that should be made explicit, as “leaking information” sounds like something which would make any reported results invalid, rather than just being something that happens during training and hurts performance. [non-critical change]
>
> Thank you for pointing this out. Your understanding is correct. Doing things this way is necessary for good generalization, but that it would not actually be incorrect to simply train on the labels as features. LaF may induce a shortcut of copying the LaF $h_{v, d+1:}^{(0)}$ to the prediction and degrade the generalization performance. We introduce Eq. (6) to prevent this. We have added a clarification to the text.
>
> > These are reasonable experiments, which are suggestive evidence for the benefits of TF-GNNs, but I think the claims made in the text are too strong given the experiments. I would ask the authors to simply add some small amount of hedging and qualifying language to acknowledge the more limited scope of the experiments, which should not be difficult. I have nonetheless answered "Yes" to the claims and evidence question on the review form, because I think these changes should be pretty small [critical change].
>
> We have changed the wording to a more rigorous one. Specifically, we used “TFGNNs are more robust to feature noise” in the conclusion previously, but have changed it to “TFGNNs are more robust to i.i.d. Gaussian noise to the node features” to match the experimental setup.

---

### Decision · Action_Editor_Y6pT · 2024-08-07

**Recommendation:** Accept as is

**Comment:**

The paper is interesting to the GNN community, and proposes terminology and mapping of existing and new ideas that allow the training-free approach of Graph Neural Networks. There is a clear consensus among the reviewers that the paper should be accepted.

**Audience:**

All reviewers agree that the paper presented interesting concepts and results and have audience within TMLR readers.

**Claims And Evidence:**

There is a consensus that the authors have addressed the concerns raised by the reviewers, and that the experiments and theoretical sections are convincing and serve as clear evidence of the claimed contributions.